# The Effect of Treatment-Associated Mutations on HIV Replication and Transmission Cycles

**DOI:** 10.3390/v15010107

**Published:** 2022-12-30

**Authors:** Madison M. Johnson, Carson Everest Jones, Daniel N. Clark

**Affiliations:** Department of Microbiology, Weber State University, Ogden, UT 84408, USA

**Keywords:** HIV-1, mutation, drug resistance, vaccines

## Abstract

HIV/AIDS mortality has been decreasing over the last decade. While promising, this decrease correlated directly with increased use of antiretroviral drugs. As a natural consequence of its high mutation rate, treatments provide selection pressure that promotes the natural selection of escape mutants. Individuals may acquire drug-naive strains, or those that have already mutated due to treatment. Even within a host, mutation affects HIV tropism, where initial infection begins with R5-tropic virus, but the clinical transition to AIDS correlates with mutations that lead to an X4-tropic switch. Furthermore, the high mutation rate of HIV has spelled failure for all attempts at an effective vaccine. Pre-exposure drugs are currently the most effective drug-based preventatives, but their effectiveness is also threatened by viral mutation. From attachment and entry to assembly and release, the steps in the replication cycle are also discussed to describe the drug mechanisms and mutations that arise due to those drugs. Revealing the patterns of HIV-1 mutations, their effects, and the coordinated attempt to understand and control them will lead to effective use of current preventative measures and treatment options, as well as the development of new ones.

## 1. Introduction

Human immunodeficiency virus (HIV) is known to exist within a host as a population of quasispecies. There will be a dominant “wild-type” version, but due to HIV’s high error rate, many subpopulations will be produced with small changes at the RNA and/or amino-acid level. There may be fitness costs associated with these changes, but they may also be advantageous. This blood-borne and sexually transmitted virus multiplies so prolifically and with enough mutations that it naturally evolves to meet its needs.

In one sense, HIV is a parasite of individual cells. During the replication cycle, each viral protein has characteristic mutations that lead to escape during antiretroviral (ARV) treatment, and each antiviral drug is associated with specific hotspots where resistance mutations occur [1,2,3,4,5,6,7]. In another sense, HIV is a parasite of humans within a population. These mutations can be transmitted from one individual to another, creating another cause of concern by enhancing HIV drug resistance and lack of vaccine efficacy [1].

This ability to adapt to host cells and to the human population occurs in the face of ARV therapies and the actions of the human immune system [1,8,9]. Man-made drugs and the elegant defense provided through immune system reactions are thus far no match for HIV, which has never been cured, although rare cases of apparent clearance and natural resistance do exist and may inform future studies. Successful vaccine development is also a current hope that may become a future reality.

This review aims to explore the effects of treatment-associated mutation on both the host-to-host transmission cycle and the cell-to-cell replication cycle. The primary focus is on HIV-1, however, most applies to HIV-2 as well. We will explain the key features that will one day lead to solutions to these current issues.

## 2. Transmission Cycle

HIV is highly transmissible through human-to-human sexual activity, vertical transmission from a pregnant or breastfeeding mother to her child, and via needle sharing and other blood exposures. Beyond abstinence, condom use, and access to clean needles, several drug interventions have attempted to stem the tide of HIV transmission. These drugs do not provide a cure, but can decrease replication and serum viral load to undetectable levels. This slows the progression to AIDS and provides better living conditions for those infected, while also decreasing transmission to others. Approaches such as prophylactic treatment of the uninfected, novel vaccine trials, and continued drug development may yield further global effects.

Currently, the state of the art is correct and consistent antiretroviral drug treatment, however HIV is highly mutable and prone to drug escape. Mutation is also the reason that every vaccine trial has failed. As the virus mutates, new battles must be fought against acquired and transmitted drug resistance. Subtle mutational changes also control viral tropism, which governs spread from host to host and clinical progression within a host. Until a cure or effective vaccine is found, mutation of HIV must be understood as the great controlling factor behind the failure of all current treatments and preventatives.

### 2.1. Acquired and Transmitted Drug Resistance

HIV-1 drug-resistance mutations can be acquired in response to the initiating antiretroviral treatment. These mutations can then be directly transmitted when the virus infects another person. The increased use of antiretroviral therapy has contributed to a steady decline of advanced infection, but also to a steady increase of emerging resistance mutations. Transmitted drug resistance (TDR) usually occurs before treatment in newly infected individuals, whereas acquired drug resistance (ADR) occurs after treatment infection with a drug-naive strain (Figure 1). Both tend to be more prominent in higher income countries due to the increased use of antiretroviral drugs [1].

Acquired drug resistance is the most common type of drug resistance, usually occurring when a patient reaches virological failure (Figure 1, yellow). Several factors may contribute to a patient developing ADR, with the most important factor being the patient’s lack of adherence [1]. Infrequent viral load testing can result in late detection of virological failure, usually after the drug resistant mutations have successful replication [10]. Non-adherence to treatment typically correlates with virological failure due to drug resistant mutations. Strictly following the current treatment plan will reduce virological escape because the mutated population will be sufficiently suppressed. if drug-resistant mutations are present with virological failure for the first time in an individual, a change in ARV treatment may have better outcomes [1]. Another possible reason for acquired drug resistance is a variation in how the drug is broken down, absorbed, used by the body, and removed from the body. These patient-specific pharmacokinetics may be due to their metabolism, diet, or other factors [11]. Non-adherence, variable drug effectiveness, and other unexplained factors can result in a fluctuating viral load that perpetuates the mutations that lead to drug resistance.

Transmitted drug resistance (TDR) occurs when an uninfected person acquires a strain of HIV that is already resistant to antiretroviral drugs (Figure 1, blue). The ultimate source of transmitted drug resistance is always from another person who has undergone treatment and has an acquired-drug-resistant virus. However, if rounds of “onward transmission” occur with a drug resistant mutant, these strains are more likely to have mutations with reduced viral fitness [1]. The replication of a low-fitness-cost mutation is favorable for the virus when compared to high fitness cost: the lower the fitness cost of a mutation, the more likely it will remain over time, even after those drugs are not used for a patient’s treatment.

On a similar note, mutants with high fitness cost can still be transmitted, especially from treatment-experienced patients [12]. These low abundance variants may remain over time until treatment is initiated in the new host, whereupon these resistant strains are selected for quickly, despite the fitness cost. These low abundance variants may be difficult to detect, but can have strong clinical effects [13]. For example, the common K65R mutation may exist at low levels in TDR cases. These pre-existing K65R mutants would then expand to become the dominant variant in a host initiating treatment, such as with tenofovir [14].

With recently infected individuals, the rates of TDR are much higher due to this replicative advantage. If viral fitness is too low when acquired by a new host not undergoing treatment, the virus tends to revert to a more wild-type genome as it is a much more viable form of the virus [15]. The increased use of NNRTI drugs as first-line treatments are correlated with the increased occurrence of TDR. TDR in a population could reduce the effectiveness of first-line treatments at a faster rate than ADR. While both forms of drug resistance are of concern, TDR has the potential to drastically change treatment options in a population and strengthen the mutation’s influence [16,17].

The current ideal would be to skew the transmission cycle toward only acquired drug resistance. This will be accomplished through a combination of the many prevention methods recommended, adherence to drug regimens, surveillance, and genotypic and phenotypic testing.

### 2.2. Drug Resistance Testing

Drug resistance testing is necessary to identify the mutations, properly treat the patient, and prevent further drug-resistant mutations. After viral escape, the source of drug resistance can be identified with genotypic testing of HIV reverse transcriptase, integrase, or protease sequences.

When either acquired or transmitted mutation happens, genotypic testing uses Sanger sequencing to identify specific mutations, and to determine an alternative treatment plan. Due to unreliable detection of low-abundance variants, next generation genotype sequencing is gaining popularity, becoming more readily available in terms of cost and accessibility, mostly in developed countries [18,19]. This type of testing requires nucleic acid extraction, PCR amplification, library preparation, sequencing, and data analysis.

Once data is obtained, test interpretation begins. The basic process includes comparing a patient’s resistance mutation to known patterns, and then changing the drug regimen if necessary. Early experiments first discovered these resistance patterns in the genome of HIV [20], and then tested outcomes based on switching to new therapy versus placebo [21,22]. However, not all mutations require a change of regimen, for example when a single component exhibits only low-level resistance, but the other components of the drug regimen remain active.

It is clear that mutation patterns in drug-induced cases are nonpolymorphic—they occur in defined patterns not seen in treatment naive patients, who have more polymorphic natural mutations. It can thus be assumed that if a patient exhibits virological escape (increased serum viral titers) and common drug-resistance mutations are present, it is directly due to the currently prescribed drug. A change in drug regimen may be indicated.

The Stanford HIV Drug Resistance Database (HIVDB, http://hivdb.stanford.edu, accessed on 1 November 2022) [4] allows input of user data and displays penalty scores that should be used to guide clinical decisions. By comparing a patient’s sequence results to databases of mutation patterns, other drug options can be considered. Each potential drug is scored for its susceptibility or resistance to a patient’s drug-resistant strain. 

In contrast to genotypic testing, phenotypic testing is used to observe the replicative abilities of an individual’s HIV in the presence of different ARV drugs by molecular cloning of PCR-amplified fragments from clinical samples into cell culture infection systems. Then, exposure to a variety of ARVs can determine the effectiveness of a drug on the individual’s strain. This provides important information to guide an effective treatment regimen, but phenotypic testing is reserved for drug research or perplexing cases [23].

While these drug resistance testing methods are feasible in developed countries, developing countries do not have the same access. Due to the monetary boundary, developing countries will empirically treat individuals that show virological failure instead of using data-based treatment. There are guidelines set by the WHO on drug-resistant treatment strategies for low-income areas [1].

### 2.3. Tropism and Its Effects on Transmission and Disease Status

Important to the idea that mutation affects transmission is HIV tropism, which defines its transmissibility and clinical progress. Successful entry of HIV into the host cell is dependent on two molecular contact points. The first point of attachment is the glycoprotein 120 (gp120) with the host cell’s CD4 receptor, prompting a conformational change of the glycoprotein. The gp120 conformational change exposes a second binding site on its V3 region that attaches to the host CCR5 or CXCR4 coreceptor. The secondary binding then activates the glycoprotein 41 (gp41) to pierce the host cell membrane and enter the cell [24]. The specific amino acid sequence of the gp120 V3 region determines the coreceptor used in attachment [25]. R5-tropic virus will preferentially use the CCR5 coreceptor, while the X4-tropic virus will preferentially use the CXCR4 coreceptor. The virus is termed dual-tropic if it can use either of the coreceptors present [26]. Understanding the specific viral tropism can enhance treatment options and control advanced infection.

Due to its direct interaction with a coreceptor, the V3 region of gp120 is typically sequenced when analyzing the tropic genotype of a patient’s HIV strain. The CXCR4-tropic strains usually have a more positive charge, higher genetic diversity, and a longer sequence. After genotyping, the “11/25 rule” is a common guideline to determine tropism. If the V3 loop mutates to a positively charged amino acid in position 11 or 25, an CXCR4-tropism is suspected [27]. Because most testing simply identifies coreceptor tropism, one group studied the genetic mutations that are associated with tropic switch. Six pairs of single-amino acid mutational patterns and three pairs of two-amino acid mutational patterns were found in gp120′s V3 loop that correlated with a higher likelihood for a tropic switch to the CXCR4 coreceptor [28]. Another group used an algorithm for tropic testing and noted that the stronger the prediction that a strain was truly R5 tropic, the more likely it would stay R5 tropic over time. No other predictors were associated with a tropic switch [29]. Tropism tests should be performed before adding treatment to a patient’s regimen, especially CCR5 entry inhibitor drugs.

There are patterns in the host-specific viral population that are centered around tropism (Figure 2). R5-tropic strains are favored for transmission of the virus and are typically found at the early stages of infection. X4-tropic strains are commonly found at the later stages of infection, but may also be present early on in significantly smaller numbers (Figure 2A). Around 50% of subtype B patients show a late stage dominance of X4-tropic strains [30]. There is a widely accepted association between X4-tropic viral emergence and disease progression, however the exact correlation is still undetermined–it could be a cause or an effect [31]. One hypothesis suggests that there is an active selection of R5-tropic viruses. It also suggests that the tropic switch is due to the virus evolving throughout the course of disease [30]. Another hypothesis suggests that memory T cells with CCR5 coreceptors are more efficiently activated, promoting replication of R5-tropic strains. Naive T cells with the CXCR4 coreceptor exhibit comparatively less HIV replication, but as memory T cell counts fall, X4-tropic strains become the favored type [32,33]. Yet, another hypothesis suggests that the immune system responds to X4-tropic strains better than R5-tropic strains. As the patient’s immune function plummets, X4-tropic strains are now at an advantage [34,35]. One study found a causal relationship between high levels of CD4 T-cell activation markers at the early stages of infection and the later appearance of X4-tropic strains [31]. Regardless of the mechanistic details, the pattern is early infection with only R5-tropic HIV, and a switch to more X4 tropism during clinical progression toward AIDS.

Our understanding of CCR5 tropism is also based on the well-known CCR5^Δ32^ mutation carried by around 10% of humans of European ethnicity, with around 1% being homozygous for the deletion. Heterozygous CCR5^+/Δ32^ carriers show slowed progression and lower viral load [36,37]. Homozygous CCR5^Δ32/Δ32^ individuals are protected from HIV-1 infection due to the inability to attach to CCR5 coreceptor, despite their cells still expressing CXCR4. One of the only successful HIV cures involved this mutation, the so-called Berlin patient. This individual was receiving treatment for both HIV-1 and advanced leukemia. After receiving a bone marrow stem cell transplant from a donor who had the CCR5^Δ32/Δ32^ mutation, the patient’s HIV was eradicated (Figure 2B). Despite the ablation of host R5-containing cells, any X4 virus remaining in the host was apparently unable to infect the newly donated cells [38].

CCR5 inhibitor-based treatment, like Maraviroc, blocks R5-tropic entry. However, this selective pressure may provide opportunity for X4-tropic entry and replication. Pre-existing X4-tropic strains have been shown to replicate rapidly when the patient is treated with R5 blocking drugs like maraviroc (Figure 2C). Once treatment is stopped, the dominant viral population reverts back to R5-tropic strains [31,39,40,41]. In most dual-tropic variants, the X4-tropic aspect is favored due to higher replication fitness. However, X4 strains or dual-tropic variants usually make up a small portion of the viral population when compared to R5 strains. When a patient is receiving maraviroc treatment, X4-tropic and dual-tropic strains have the selective advantage, promoting replication [42]. The high mutation rate of HIV is the fundamental source of all of these described tropism adaptations.

### 2.4. Vaccine and Prevention Strategies

Stopping HIV infection before it begins would be ideal. Several strategies are discussed in this section. During vaccine trials, the human immune system sometimes creates a strongly protective antibody response when it produces broadly neutralizing antibodies (bNAbs). Passive antibody transfers of bNAbs have also been attempted. Pre-exposure prophylaxis (PrEP) using treatment drugs in pre-exposure settings will likely prevent many new infections. Novel future technologies include mRNA vaccines that may encourage both the production of bNAbs and a strong T cell response, and CRISPR gene editing that may reduce or eliminate latent reservoirs of cells carrying integrated HIV genomes.

#### 2.4.1. Vaccine Trials and bNAbs

All vaccine trials have failed thus far due to immune-escape mutations in HIV. However, through trial and error, the path towards a successful vaccine or treatment is becoming clearer. Broadly neutralizing antibodies (bNAbs) bind to gp120 or gp41 (Figure 3) and have the potential to match the mutational speed of HIV, maintaining inhibition of attachment to CD4 receptors. Due to this ability, the production of bNAbs is the current gold-standard in HIV vaccinology, although this standard has not yet been reached at a large scale.

Over the course of HIV-1 infections, individuals will develop neutralizing antibodies that work with only a narrow range of strains. Broadly neutralizing antibodies are highly potent antibodies that work by binding to one of several key sites on gp160 (Figure 3). Affinity maturation (due to B cells’ somatic hypermutation) yields antibodies that continually adjust to changing antigens resulting from HIV-1′s prolific replication errors [48]. First generation bNAbs were initially isolated from blood and found to bind at gp120′s V3 loop, the CD4 binding site (CD4bs), and gp41′s membrane-proximal external region (MPER). The new generation of laboratory-generated monoclonal antibodies bind at new sites–the gp120-gp41 interface, the “silent face” of gp120, and the V1V2 apex–in addition to the sites bound by the first generation. The new generation also has a monumental increase in potency and coverage [49].

Two antibody-mediated prevention trials hoped to find success in passive administration of a broadly neutralizing monoclonal antibody. Intravenously administered VRC01 was delivered in increments over the course of 20 months in trials HVTN704 and HVTN703 (Figure 4). Unfortunately, the antibody did not sufficiently prevent HIV-1 infection in participants. The majority of HIV strains in newly infected individuals were found to be insensitive to the VRC01 monoclonal antibody. When this single antibody was administered, HIV was also able to quickly outmaneuver it. With each injection, selection pressure led to a high amount of VRC01-resistant strains [50].

Vaccine trials (Figure 4) will be discussed briefly, emphasizing that the mutation of HIV promotes vaccine failure or escape. The RV144 clinical trial used a canarypox virus-based vector encoded with HIV-1 subtype B’s Gag protein and Pro protein inserts along with the Env protein from the subtype circulating recombinant form (CRF) 01_AE, administered at zero, one, three, and six months. At three and six months, the trial also administered gp120 proteins from subtype B (isolate MN) and CRF01_AE (isolate A244). The trial, conducted between 2003 and 2006 with 16,402 patients, resulted in moderate vaccine efficacy (31%), leading to the end of the trial [51]. A follow-up study was done to evaluate the long-term effects of the RV144 vaccine clinical trials in the placebo group compared to the vaccine group after participants became infected with HIV-1. In the placebo group it showed that there were more effective responses to IgG3 and gp120 specificity. This long-term reaction juxtaposed that of the vaccination group, where there was decreased IgG3 binding to Env, increased IgG2 and IgG4 binding to Env, and increased V1V2-specific binding by IgG1. The vaccination group also saw a lack of bNAbs developed after infection but an increase in Fc-mediated effector functions. This indicates that B cells were primed to have very specific antibody responses with prevalence given to V2 antibodies which lacked neutralizing capabilities [59]. With the inevitable mutations of HIV-1, hyper-specific binding sites, as seen with vaccination but not natural infection, are not effective or efficient in terms of immune response. 

The specificity seen in vaccinated participants post-infection hindered the development of bNAbs and also supported a sieve analysis that showed V2-specific responses to vaccination. The analysis looked at breakthrough HIV-1 genome sequences from the vaccine group and the placebo group. The 936 different genomes showed a significant distinction between the two groups’ immune responses associated with amino acid positions 169 and 181 (Figure 3). The residue at 169, a common target for bNAbs and RV144-derived antibodies, had the greatest vaccine-induced immune response when patients’ viruses matched that of the vaccine they received. Amino acid site 181 showed the greatest vaccine-induced immune responses when viruses were different from that of the vaccine, suggesting a limitation due to the vaccine [44]. 

In the RV305 trial, there was a promising reaction to boosters in uninfected participants that were previously vaccinated in the RV144 trial. Receiving a booster eleven years after receiving the treatments in the RV144 trial showed an increase in somatic hypermutation. This is significant to develop bNAbs and respond to the rampant mutations in the HIV-1 virus [60]. The RV144 vaccine trial showed that vaccine-induced immune responses work well when specifically binding to V2 sites, but HIV-1 mutations still pose a problem with vaccine efficacy and subsequent bNAb development. The commonly supported solution to this is increased somatic hypermutation of the HIV-specific antibody repertoire, matching bNAb mutations with HIV-1 mutations.

The STEP trial recruited 3000 HIV-1 negative subjects in the US and Latin America. Subjects received three injections of three rAD5 vectors with each vector expressing Ad5-gag, Ad5-pol, and Ad5-nef [61]. The goal of the vaccine was to induce T cell immunity to HIV for a reduced viral load in breakthrough infections. The trial was stopped midway because there was not adequate vaccine efficacy to prevent HIV infection. Furthermore, the trial did not show a reduced viral load post-infection compared to placebo for those that became infected during the trial. Out of 741 vaccinated participants, 24 presented with breakthrough HIV-1 infections, compared to 21 out of 762 in the placebo group. Results from this study suggested that some vaccinated participants had pre-existing high levels of antibodies against adenovirus serotype 5 before vaccination. It was concluded that participants were at a higher probability of HIV-1 infection because of those pre-existing antibodies [62]. This trial shows that using a vector that the body has not been exposed to may be necessary for an HIV-1 vector vaccine. As with most vaccine attempts, this early trial showed that the static protein antigens the immune system targets become an ineffective representation of the mutable protein antigens presented by the changing HIV-1 virus. 

HVTN 702 is a vaccine trial that utilized a recombinant canarypox vector with subtype C envelope ALVAC-HIV and a subtype C bivalent gp120 protein adjuvanted by MF59 [57]. The addition of MF59 to create stronger binding and higher concentrations of neutralizing antibodies [63] was thought to enhance antibody development where the RV144 trial saw deficiencies. This trial in South Africa from 2016 to 2019 saw no difference in the viral load of breakthrough HIV infections between vaccinated individuals and the placebo group participants. Because this trial used subtype C, it is possible that the lacking results are due to the great genetic diversity of this subtype when compared to RV144′s use of subtype B [57].

Newer trials HVTN705 (Imbokodo) and HVTN706 (Mosaico) are expected to end within the next two years. They use a priming vaccination of Ad26.Mos4.HIV at four different time periods with boosting adjuvanted vaccinations of Clade C gp140 protein administered twice, with the Mosaico trial adding a mosaic gp140 protein. The results from these trials are eagerly awaited. It is hoped that bNAbs will be generated in a way that the virus cannot avoid through mutational escape.

#### 2.4.2. mRNA Vaccines

mRNA vaccines promote the cell-mediated immunity required to kill virally infected cells and also lead to a specific antibody response. HIV-1 mRNA vaccine research utilizes either non-amplifying mRNA vaccines or self-amplifying mRNA vaccines. Non-amplifying mRNA vaccines encode only the immunogen. They yield a finite immune activation, limited by the amount of mRNA transcripts in the vaccine. Because of this limitation, efficacy and the amount of mRNA are directly correlated. In the case of self-amplifying mRNA vaccines, both the immunogen and the parts needed to replicate it are encoded, but without creating infective viral particles. They require a smaller dose and are more effective at enhancing immune responses [64,65]. A self-amplifying RNA vaccine study in nonhuman primates found evidence of eliciting both cellular and humoral immune responses, even at low doses [66]. Due to the success of messenger RNA vaccine technology with SARS-CoV-2, there is promise for a successful HIV-1 mRNA vaccine. In 2021, a Phase 1 study for an HIV-1 mRNA vaccine was announced [67]. The hypothesis is that this method will induce the development of bNAbs in the participant, which are the only known hope to combat HIV-1′s mutations.

#### 2.4.3. Preventative and Post-Exposure Treatments

Classic prevention strategies include avoiding exposure and effective vaccines. In the absence of these, the novel ability to use preventative medications is an important step to reduce risk of infection. Several recent drugs have been introduced. Preexposure prophylaxis for HIV provides at-risk groups with a preventative safety net. It has significantly lowered the probability of being infected with HIV among these groups. There are currently two drugs used for PrEP, emtricitabine and tenofovir. 

The most common mutations seen with HIV also contribute to PrEP resistance. Resistance to tenofovir (M184I/V mutation) and resistance to emtricitabine (K65R mutation) have shown to be the most common drug-resistant mutations in HIV-1 infections [68]. A study was done to see which PrEP-associated mutations were present before and after breakthrough HIV-1 infection. Four of the cases suggest resistance mutations K65R, K70E, and/or M184I/V. These participants had all started taking PrEP while they had an acute HIV infection. Acute infections have a high replication rate and therefore high rates of mutations due to the notorious errors made by HIV. The ultimate findings from this study reported that at six, twelve, and twenty-four months after participants stopped taking PrEP, all PrEP-related mutations were undetectable. The development of resistance to PrEP is low but can occur if PrEP is taken while acutely infected. One patient in the study had no history of PrEP, but showed M184 mutations in their viral load, suggesting transmitted drug resistance. In the case of acquired resistance, cessation of PrEP is necessary to remove selection pressure and stop the replication of drug-resistant mutations [69]. 

Consistently following the recommended medication usage is necessary to prevent infection with HIV [69]. Traditionally, PrEP is taken daily. However, there is a new secondary strategy for the oral PrEP pill called 2-1-1 dosing. Preference to this strategy is seen in people with low rates of sexual encounters, people seeking reduction in cost, people who forget to stick to a daily regimen, and people willing to reduce side effects. This design incorporates taking two tablets 2–24 h before a sexual encounter, followed by two tablets taken separately between 24–48 h after the initial dosing of two tablets [70,71]. Taking PrEP daily could potentially contribute to drug resistance, whereas the 2-1-1 dosing and other new PrEP strategies could prevent that possibility [72]. 

An alternative version of PrEP, Cabenuva (cabotegravir extended-release injectable suspension; rilpivirine extended-release injectable suspension), was approved by the US FDA in 2021. This injectable medication is only administered once a month. Efficacy trials including 591 participants found no difference in the effectiveness of daily, oral PrEP and the once-a-month injectable Cabenuva. However, six participants from the trial had breakthrough infections that presented with drug resistance mutations. Similarly to PrEP, once treatment with Cabenuva was replaced with a treatment medication, these mutations became undetectable in the participants. Because missed doses are less likely to happen with this once-a-month alternative to PrEP, drug-resistance mutations are suspected to decline [73]. 

Similar to PrEP, post-exposure prophylaxis (PEP) is another form of preventative treatment for people who have been potentially exposed to HIV, especially in the workplace. PEP includes tenofovir and emtricitabine treatment, as well as raltegravir or dolutegravir for 28 days. Similar to PreP, drug resistance is seen when the treatment regimen is not followed and when the individual is infected with an already drug-resistant virus [74].

#### 2.4.4. CRISPR

Although we can suppress replication, infected cells contain the integrated viral genome. This provirus must be removed to cure the infection from the human body. The use of clustered regularly interspaced short palindromic repeat (CRISPR) technology can be adapted to target these latent reservoirs, suppress replication, and prevent transmission [75]. Inactivation of the proviral DNA in latent reservoirs with CRISPR could lead to an HIV-1 cure. In studies evaluating the efficacy of CRISPR/Cas9 edits of integrated HIV genomic targets, sustainable viral inhibition was achieved when the target sequence was well conserved in HIV-1. However, because of HIV-1′s rapid evolution, escape mutations at the cleavage site occur in response to Cas9 [76,77]. When targeting human genes instead of viral genes, CRISPR/Cas9 inactivation of CCR5 is an attractive strategy, however this could potentially lead to Env mutations that allow the HIV-1 strain to use CXCR4 instead. CRISPR/Cas13a cleaves RNA in order to inhibit new virus production and latent reservoirs. Cas13a was found to reduce the amount of newly synthesized viral RNA and dismantled viral RNA within capsids [78]. However, because Cas13a works only on RNA, not DNA, a theoretical HIV cure cannot be found with Cas13a alone. Because these findings are only seen in a laboratory setting, more research in humans is needed to evaluate efficacy. 

Preventatives such as a safe, highly effective, and accessible HIV vaccine will be significant contributors to the interruption of the chain of transmission of HIV. Well-conceived HIV immunization strategies could reach populations where other interventions are not sufficiently effective. In combination with antiretroviral therapies, preventatives will decrease the cost of treatment and increase long-term efficacy. To be effective, a robust immune reaction, such as the production of bNAbs, is required. Preventatives face the challenge of mutation, the same challenge that faces current treatments, as discussed below.

## 3. Replication Cycle

In addition to the person-to-person transmission discussed above, HIV transmits from cell to cell. The replication cycle of HIV in individual cells consists of several main steps: attachment, entry, reverse transcription, integration, biosynthesis, assembly, release, and maturation due to protease activity. Blocking some basic steps can be accomplished with therapeutic drugs, however, HIV is a rapidly mutating virus. Therefore, HIV naturally develops mutations to prevent drug inhibition. Although a widely-use centralized database of mutations exists that collects mutation data for research and clinical use [1,2,3,4,5,6,7], not all cases are treated, and not all treatment-escape cases are sequenced to determine why treatment failed. Information is a powerful tool for staying ahead of treatment associated mutations. Replication cycle steps will be discussed and related to drug inhibitors and drug-escape mutations (Figure 5). 

### 3.1. Attachment and Entry

The attachment of the virion is initiated by the fusion protein, gp160 (Figure 5, step 1). This protein is trimerized and then cleaved, producing three copies of each gp120/gp41 subunit. The HIV particle will attach to a host cell’s CD4 receptor with gp120, prompting a conformational change of this glycoprotein. This change enables gp120 to also bind to one of two coreceptors, CCR5 or CXCR4. Next, the N-terminal fusion peptide gp41 moves and pierces the host cell membrane, exposing gp41′s C-terminal peptide. From here, three gp41 molecules fold to bring both fusion peptides together to form a six-helix bundle that brings the membranes even closer, creating a fusion pore through which the virion reaches the host cytoplasm. The cone-shaped capsid then releases the viral genome [24,79,80,81].

There are currently four FDA approved drugs that are involved in the attachment and entry step; Enfuvirtide, Maraviroc, Fostemsavir, and Ibalizumab (Table 1). Enfuvirtide (T20) was the first FDA approved peptide-based HIV entry inhibitor. Starting in 2003, this fusion inhibitor was used by patients faced with virologic failure. It acts like the gp41 C-terminal peptide to encourage N-terminal peptide binding, but then blocks the folding mechanism. This prevents the formation of the six-helix bundle and subsequent fusion pore. Enfuvirtide is currently only recommended for patients facing multidrug resistance and is conservatively prescribed [82,83]. Maraviroc (MVC) is currently the only approved CCR5 antagonist that inhibits the secondary coreceptor binding step needed for viral entry. R5-tropic strains treated with MVC are unable to infect the cells, similar to people with the CCR5^Δ32/Δ32^ mutation. Inhibition is suspected to only work if the individual does not have active X4-tropic or dual-tropic viruses [84]. Due to this, a viral tropism test is necessary prior to treatment [85]. A recently approved inhibitor, Fostemsavir, competes with the CD4 receptor and prevents initial attachment of the virus. There is currently no known cross-resistance with other ART treatments, making it highly suitable for patients with multidrug resistance [86]. Additionally, affecting entry, Ibalizumab was FDA approved in 2018 for multi-drug resistant HIV and used in combination with ART. Ibalizumab is a humanized monoclonal antibody that does not prevent the initial attachment of the CD4 receptor, but does interact with the CD4-gp41 complex, preventing coreceptor engagement [86].

Patterns of drug resistance for these novel attachment and entry inhibitors are emerging (Figure 6; Table 2). Enfuvirtide mutations are generally found in the highly conserved gp41 [87]. There are mutations that can increase CD4 cell count or decrease CD4 cell count with no significant changes to viremia [88]. Drug-resistant mutations that are associated with Maraviroc have generally been found to cause a strain-specific change within the V3 loop of gp120 [89]. In vitro research also found resistance mutations in the C4 region of gp120 from a subtype A virus. The results suggest that an N425K mutation may impact CD4 interactions, reducing susceptibility to MVC. In association with the high replicative fitness cost of N425K, a common group of mutations are thought to provide stability to the structure; E33G, R117Q, Q290K, L/G396V, and D461E [90]. Other mutations in the V3 loop are necessary for extensive replication, necessary for competitive resistance, and enhance viral fitness. T199K mutation is uniquely found in the C2 region and enhances viral fitness [91]. Fostemsavir mutations are found in gp120 and associated with reduced drug susceptibility such as S375 and M426. Generally, any non-M genotype strain of HIV-1 has natural resistance to the drug [92,93,94]. The most common substitution mutation found was S375T, however, this mutation does not have a strong effect on resistance [93,94]. Deletion of a potential N-linked glycosylation site (PNGS) in gp120′s V5 loop is most often attributed to Ibalizumab resistance. When paired with the absent site, the length of the V2 loop can cause varying degrees of resistance strength. Two mutations have been found to disrupt the PNGS and are thought to be associated with this deletion [95].

### 3.2. Reverse Transcription

Reverse transcription is a unique process that occurs in the retroviridae family of viruses, where a ssRNA genome becomes a dsDNA copy (Figure 5, step 2). In HIV, reverse transcriptase takes place in newly infected cells. The two main enzymatic activities, found in a single enzyme, are DNA polymerase activity (due to the RT domain) and RNase H activity (due to the RNase H domain). The RT domain has 3 subdomains named using a hand analogy. The template slides through the palm, which contains the enzyme’s catalytic center, guided by the fingers and thumb at either side (Figure 7). A heterodimer of RT (p51) and RT plus RNase H (p66) form the holoenzyme, where p51 plays a role in structural support and p66 performs all catalytic functions [97].

DNA polymerase’s function is to copy either an RNA or a DNA template. RNase H’s function is to degrade RNA that is part of an RNA-DNA duplex. During reverse transcription (in the case of HIV-1) the host tRNA primer is Lys3. The template viral RNA contains long terminal repeats (LTR), whose repeated segments allow for several strand transfer reactions. A partial negative strand is made first at the 5′ LTR, then translocated to the 3′ LTR for full strand synthesis. The RNase H activity follows, degrading the 5′ end of the viral RNA and exposing the new negative strand DNA. DNA synthesis continues down the length of the genome, but the polypurine tract is resistant to RNase H and remains intact to act as a primer for positive strand DNA synthesis. The reverse transcriptase then copies the negative strand as well as the first eighteen nucleotides of the tRNA host primer. Reverse transcriptase then cleaves the tRNA one nucleotide away from the 3′ end leaving a single A ribonucleotide at the 5′ end of the negative strand. A full length dsDNA genome copy is thus made [99].

There are currently two types of drugs that target reverse transcription, nucleoside reverse transcriptase inhibitors (NRTI) and non-nucleoside reverse transcriptase inhibitors (NNRTI), with FDA-approved inhibitors listed in Table 3. The NRTIs are competitive inhibitors that function as prodrugs, after metabolizing the drug (usually by phosphorylation to a triphosphate nucleotide) it becomes a pharmacologically active drug. The NRTIs are then incorporated into the viral DNA using reverse transcription. The usual mechanism is chain termination, as NRTIs tend to lack a 3′ hydroxyl group. The names given to NRTI mutations can either be nucleoside/nucleotide associated mutations (NAMs) or thymidine analog mutations (TAMs). NNRTIs are non-competitive inhibitors that bind to the active site and form hydrophobic pockets adjacent to the active site. The mutation common with NNRTIs is a change in the amino acid sequence resulting in unfavorable binding sites for the NNRTIs. The degree to which these mutations decrease drug binding is quantified in Figure 8; the most common mutations are listed in Table 4.

### 3.3. Integration

After entry, uncoating, and reverse transcription take place, the resulting double-stranded viral DNA must be inserted into the host genome (Figure 5, step 3). Integration is a necessary step in the viral replication cycle that happens in all RNA retroviruses [100] and is catalyzed by the viral integrase enzyme (IN). IN is a 280-residue protein with three active-site amino acids (D64, D116, E152) that coordinate Mg^2+^ and allow catalytic activity, found in the core domain (Figure 9). The IN monomer also consists of an N-terminal Zn-binding domain and a C-terminal DNA binding domain, which promote the interactions and assembly of components that make integration possible.

Prior to integration in the cytoplasm, viral DNA is associated at either end with dimers of the viral integrase (IN) enzyme. The IN enzyme itself is only part of a larger complex of proteins, called the pre-integration complex, that includes viral dsDNA, IN, Vpr, and matrix proteins; host proteins BANF1 and LEDGF/p75 are also associated [106,107]. The host factor TRIM5α is known to interfere with uncoating prior to integration in some non-human primates, however, the human TRIM5α gene is not able to do so [108]. Human genes that are required for integration represent an interesting mechanism for the future of HIV replication inhibition.

The pre-integration complex forms in the cytoplasm and removes two nucleotides from only the 3′-ends, preparing these sticky ends for insertion into a host chromosome [109]. The matrix protein contains the nuclear localization signals that direct the pre-integration complex to enter the nucleus [110] where IN catalyzes the strand transfer reaction. Recombination of viral and host DNA always occurs at the same site in the viral genome, the 5′ and 3′ LTRs, but can occur almost anywhere in the host genome. There is a preference for transcriptionally active sites, likely to keep HIV gene expression active after integration [111].

Integrase strand transfer inhibitors (INSTIs) came into use in 2007 and are a mainstay in current treatment options (Table 5). They are well tolerated and commonly used in combination with NRTIs. All INSTIs have the same basic function, acting as competitive inhibitors of the active site of the integrase enzyme. Chemically, they are diketo acids with multiple ring structures that chelate away the Mg^2+^ ions, also binding to the IN active site when in complex with viral DNA [112,113]. First generation drugs RAL and EVG led to escape mutations, usually initiating at Y143, Q148, or N155 and commonly with additional mutations [114,115]. Due to lower rates of virological escape mutants, DTG and BIC are current first-line therapies in combination with NRTIs. Long-lasting injections with CAB make it a drug of choice, but mutations occurred at G140 and Q148 during clinical trials and in additional sites in vivo [116,117].

INSTI treatment-associated mutations occur most commonly in the core domain (Figure 9, light blue circle), adjacent to active-site residues. Most mutants that confer resistance are in an unstructured loop spanning positions E138 to S153, where mutations likely block drug disruptions. For example, a mechanism of resistance at Q148 has been described, where the substitution displaces a critical water molecule that helps coordinate both D116 and E152 active site residues; this in turn changes the charge and interaction with the critical Mg^2+^ ion, allowing catalysis despite the presence of inhibitors [118]. Substitutions at E138 and G140 may disrupt networks of hydrogen bonds in this area that allow escape [119]. Outside the IN core domain, mutations that confer resistance can occur in the DNA-binding domain (Figure 9, light green circle). RAL and EVG binding sites are modeled to occur near DNA-binding sites such as R263. The most common INSTI-resistance mutations are listed in Table 6.

Additionally, of note, the aromatic rings of INSTIs may interact directly with DNA nucleotides, not just protein, and disrupt the multimeric complex [119,120]. Indeed, INSTIs interact preferentially with DNA-bound complexes, implying drug effect at a complex level and not a monomer level [121]. These direct interactions with DNA have been observed specifically with DTG, where mutations in DNA nucleotides at the 3′ polypurine tract led to virological escape [122,123].

After integration, the HIV-1 genome acts as a transcriptionally active genetic element, a provirus. Expression of viral mRNA is driven by strong promoters in the long terminal repeats (LTR) at either end of the genome. Like host mRNA, HIV mRNA becomes capped and polyadenylated and contains a single start codon and stop codon. The production of these viral components will lead to release of new HIV particles.

### 3.4. Assembly and Release

Viral components will form new viral particles during assembly and release steps. Because these steps are essential, assembly and/or release inhibitors may be excellent targets for drug inhibition. However, these steps have not yet been successfully targeted by therapeutics (Figure 5, step 4). While this process is included here for a complete view of the cycle, it is likely that only natural mutations, and not drug-induced mutations, play a role in these steps. During assembly and release, the Env glycoprotein, Gag polyprotein, and Gag-Pol polyprotein, along with a variety of host factors, all converge along with the HIV genome near the cell surface and new particles are released by budding [124].

The target for Gag protein-related drugs are aimed at viral maturation and budding. Areas of promise include the small-molecule capsid inhibitors of capsid function and maturation inhibitors which inhibit the final steps of the Gag process. Bevirimat is not currently approved, but was first studied as a maturation inhibitor. This drug works by binding immature particles at Gag’s capsid-spacer peptide 1 (CA-SP1) junction, preventing maturation. Mutations were observed at this specific junction after being exposed to the drug [125]. Another studied drug, PF96, works by a similar mechanism as bevirimat. There were shared mutation sites to bevirimat, but additional mutations also found upstream from this site, at the capsid major homology region, that stabilized the virion’s structure [126].

Although these drugs are not in current use, the resulting information provides insight into the maturation process and important components of it. Drug-associated mutations will continue to be a hurdle for HIV treatment, including future assembly and release inhibitors.

### 3.5. Maturation and Protease Activity

Maturation of the virus occurs after new particles are released from an infected cell, due to the activity of protease (Figure 5, step 5). Maturation turns Gag and Gag-Pol polyproteins into functional cleaved units [124]. The resulting protein cleavage products become viral structural proteins (matrix, capsid, and nucleocapsid) and enzymatic proteins (protease, reverse transcriptase, and integrase). Protease also cleaves Nef and Vif [127,128,129]. Additionally, HIV protease cleaves host factors EIF4GI and PABP1 which skews translation in favor of HIV protein production [130]. The highest protease activity levels occur as part of particle maturation during budding, just before release from the cell [131].

Protease is a small 99-amino acid protein that forms a homodimer (Figure 10). At the interface of the dimers, several active-site residues are associated with protein targets. A beta strand forms a hinged flap region, keeping the target protein near the active site. Cleavage sites on target proteins are non-homologous and based on shape more than sequence [132], but are usually between Tyr-Pro or Phe-Pro. Drug design against protease used this knowledge to identify peptide mimics that would bind the active site competitively [133]. Protease inhibitors (PIs) were the second class of antiretroviral drugs developed (after RT inhibitors), and were first US FDA approved in the mid 1990′s, beginning the era of combination HAART therapies.

PIs bind competitively to the active site, D25-T26-G27 in HIV-1, which is also found in functionally related peptidases [135]. Drugs also interact with the larger substrate-binding cleft and the flap region (Figure 10, green residues and blue region, respectively). Currently, there are ten protease inhibitors approved by the US FDA (Table 7). Only DRV/r and ATV/r are recommended for initial use in some patients. The “/r” referred to in drug coding refers to the use of ritonavir as a pharmacological booster, where it is used in combination with other protease inhibitors since it slows their being metabolized, thus accentuating other drugs’ activity [136]. Among the three common PIs, DRV/r has the highest barrier to resistance, followed by LPV/r, and then ATV/r [1].

Years of experience with PI therapy, treatment escape, and subsequent sequencing have revealed patterns in drug resistance mutations. It is well established that PI resistance requires multiple mutations in order to stop PIs’ inhibitory functions. Most drug inhibition begins with a single mutation at the active site which reduces competitive binding (primary mutation or inhibition), and confers strong resistance only with compensatory secondary mutations that affect enzyme shape by changing hydrogen bonding and recovering fitness [137]. For example, DRV/r is susceptible to resistance when the virus exhibits a V32I non-active site substitution, but V32I happens more readily if the virus already has I54M and I84V substitutions (note that I84 is in the active site) [138].

There are 13 major mutants that confer resistance to PIs when mutated (Table 8). Seven are near the active site (D30, V32, I47, G48, I50, V82, I84) in regions that fold together to form the substrate-binding cleft (Figure 10). V82 substitutions such as threonine were shown to alter the hydrophobicity of this binding cleft [139]. With this in mind, LPV/r was specifically designed to not cause V82 mutants [140] although V82 mutation does confer resistance to LPV/r. Two additional major mutations, M46 and I54, are in the flap region (Figure 10, positions 46–56), which engages with the active site upon substrate binding. Three other resistance-associated mutations are outside the cleft and flap: L76, N88, L90. These are usually explained as compensatory secondary mutations or those that alter interactions with cleavage sites, where they bind to the so-called substrate envelope [132].

As mentioned above, cross-resistance can occur and yield treatment-associated mutations that confer resistance to multiple drugs, and this is also the case with PIs, such as LPV/r and ATV/r [141]. Cross-resistance can also work in reverse. In cross-susceptibility, other drugs work better after the virus mutates to resist initial treatment. The main example of cross-susceptibility for PIs occurs when atazanavir resistance occurs with a I50L mutation, which is usually accompanied with a fitness compensating A71V mutation [142]. This I50L/A71V mutation actually increases binding by other protease inhibitors by 2- to 10-fold, increasing their activity [143]. Although protease inhibitors are not included in initiating therapy and they are always used in combination, their continued use in the many second-line regimens contributes to the overall resistance of HIV and must continue to be monitored.

## 4. Conclusions

The disease-causing virus HIV is an agent that suffers from a disease itself-it mutates rapidly. Because of this, RNA viruses such as HIV exist as populations of quasispecies, even within a single host. Yet, HIV moves past the loss of less competent virions, because of the renewal provided by drug-resistant mutants. Therefore, our efforts to prevent and control HIV infections are outsmarted by the simplest of nature’s tools: natural selection. 

In 2021, 1.5 million people became newly infected with HIV, spreading the virus from host to host [58]. Transmission can be controlled by preventatives such as PrEP, but mutations undermine their effectiveness. Transmitted drug resistance can rapidly progress the transition to AIDS if it is not quickly suppressed by ART. Although effective treatment options exist, mutations can render them ineffective and new treatment plans must be implemented.

Several questions remain. It is known that resistance mutations are generally found in the genes that are targeted by the antiretroviral treatment, but sometimes resistance is attributed to other mutated genes that are not the specific target of drugs. Although rare, these mutations can arise and should be studied further, although this would require full genome analysis, instead of only sequencing specific regions [144]. If mutations lead to escape, perhaps too much mutation could promote lethal mutagenesis, such as may be induced by favipiravir and molnupiravir that utilize viral error catastrophe. More research should be done to address concerns of genotoxicity and carcinogenic risks to the host over a long period of time [145,146]. Means of removing latent HIV reservoirs are elusive. These reservoirs are the main reason that a complete cure for HIV has not been found. Though ART can cause undetectable viral loads, latently infected cells have a half-life of 44 months and can reactivate infection. Further research on reservoir formation, persistence, and reactivation are needed to take the next step in finding a cure [147,148,149]. 

What lessons have we learned? Multiple targets work better than single targets, as seen in combination therapies and broadly neutralizing antibodies—both of which reduce escape mutation rates. Combination therapies are the best option because drug resistant mutations arise at or near the site of drug interference. For patients with multidrug resistance, attachment and entry inhibitors are often utilized. While treatments are advancing, only perfect drug adherence would ensure complete viral suppression of drug-resistant mutants. Once a patient has acquired HIV, frequent testing and early intervention are established ways to monitor and combat drug-resistance.

However, “multiple targets” could also describe the coordinated efforts of government, pharmaceutical, scientific, and healthcare agencies, along with the community and the individual. To increase awareness, campaigns can provide education, testing, and treatment. When HIV-positive status is known, immediate access to treatment and education is critical. Monitoring overall patient health, not just HIV progression, will improve health outcomes. These testing and monitoring efforts can inform patient care for patients and their sexual or needle-sharing contacts over time. Because knowledge is power, sharing risk data, care data, and treatment best practices can be expanded and standardized. Databases of mutations and sequencing efforts have reduced time wasted on ineffective treatments in more developed countries, but individuals in low and middle-income countries lack the resources to keep up with effective treatment plans, which can lead to an increase in transmitted-drug resistance. Research will continue to support new therapeutics, preventatives, and vaccine candidates, funded by many levels of philanthropy and government investment. Addressing misinformation and healthcare mistrust will continue to be important. These multifaceted strategies can match the overwhelming speed of viral replication and the subsequent mutations. Coordinated efforts are required at many levels to achieve successful results, but will eventually lead to a cure.

## Figures and Tables

**Figure 1 viruses-15-00107-f001:**
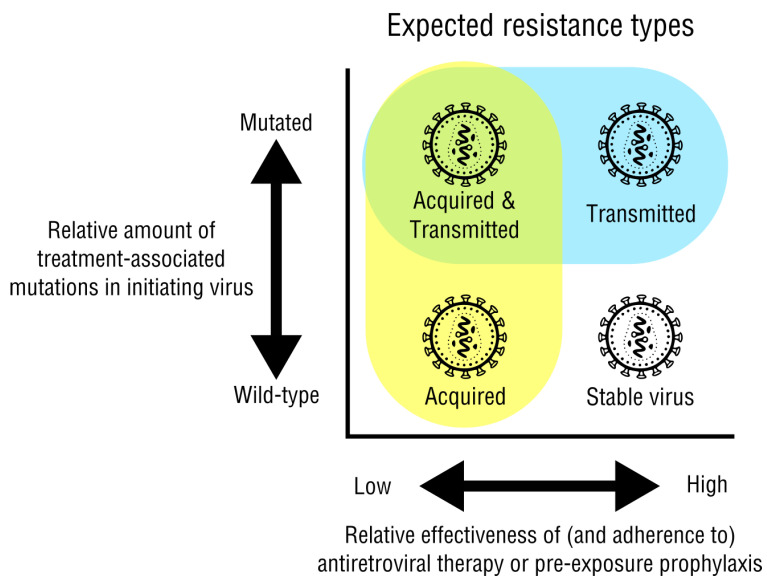
Expected resistance types due to treatment effectiveness and the characteristics of the initiating virus strain. Acquired (yellow) and transmitted (blue) resistance are shown. Viruses that infect as a wild-type are unmutated and highly susceptible to antiretroviral therapy. Ideally, when therapy is effective, this strain would not develop escape mutations and any spread to new hosts would remain susceptible to treatment (lower right). After therapy however, these strains could acquire drug-resistance mutations (ADR, lower left). Viruses that infect after acquiring drug resistance are less susceptible to treatment, potentially acquiring additional mutations if less effective drug combinations are used or adherence is inconsistent (upper left) or spreading with existing mutations to new hosts, referred to as transmitted drug resistance (TDR, upper right). Pre-exposure prophylaxis would also be most effective against wild-type viruses.

**Figure 2 viruses-15-00107-f002:**
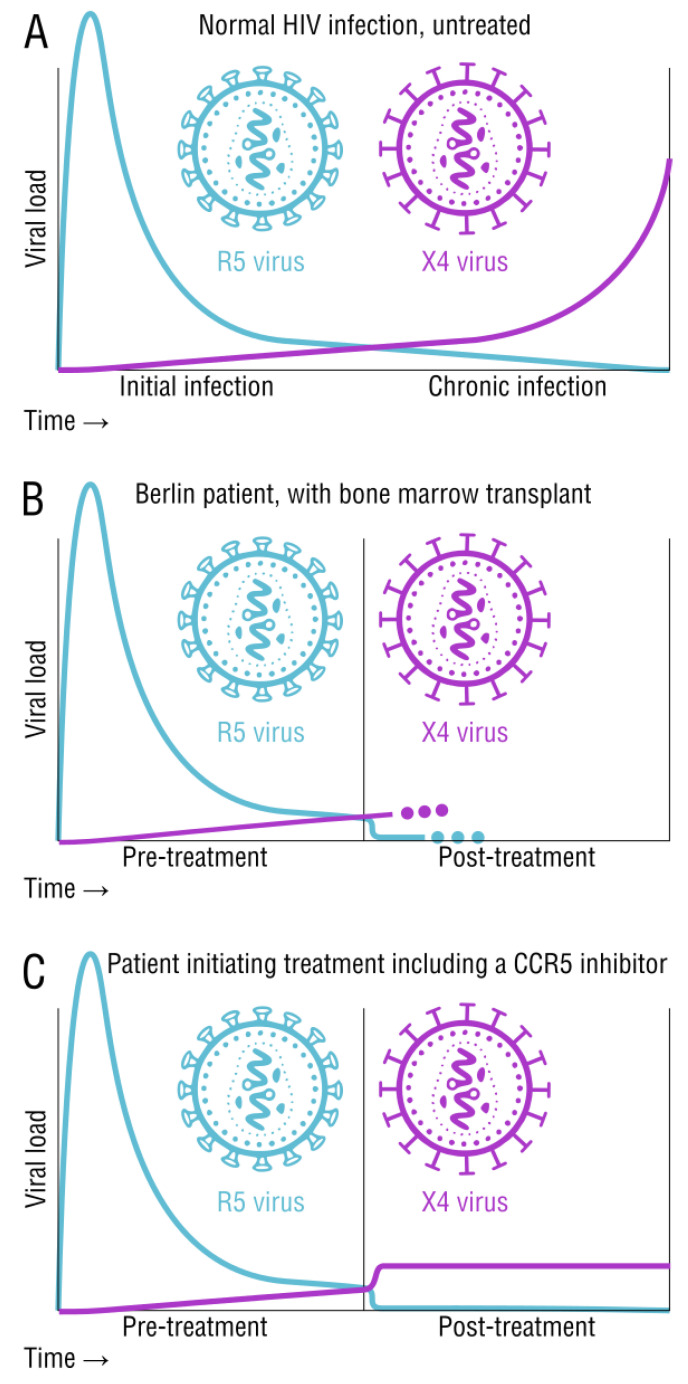
The outcome of tropic switching over time due to HIV mutation. (**A**). The normal pattern of HIV infection starts with becoming infected with the R5 variant which attacks the CCR5 receptor. The virus then mutates and becomes X4-tropic, gaining entry via the CXCR4 receptor, also correlating with the progression to AIDS. (**B**). The Berlin patient received a bone marrow transplant from a donor with the CCR5^Δ32/Δ32^ mutation, however the patient was cleared of both R5-tropic and X4-tropic HIV, a rare and unusual cure. (**C**). Treatment with CCR5 inhibitors like Maraviroc quickly induces a tropic switch to X4 variants. Viral load remains low due to other components that would be used in combination.

**Figure 3 viruses-15-00107-f003:**
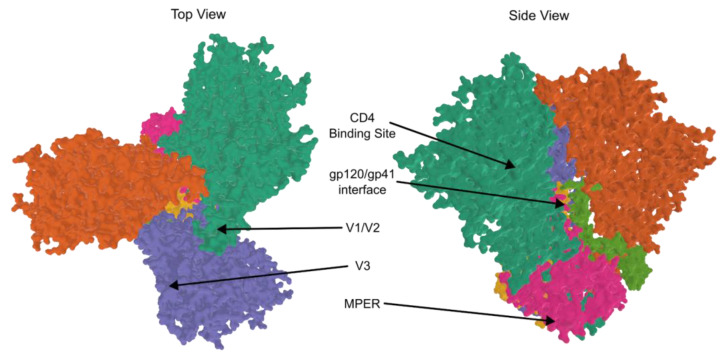
bNAb binding sites. Individual monomers are shown in orange, green, and purple and form the trimeric gp120. Yellow, pink, and neon green depict monomers within the trimeric gp41. The bNAb binding sites are indicated with arrows: the CD4 binding site, gp120/gp41 interface, V1/V2, V3, and MPER. Drawn from PDB ID: 6VRW [43,44,45,46,47].

**Figure 4 viruses-15-00107-f004:**
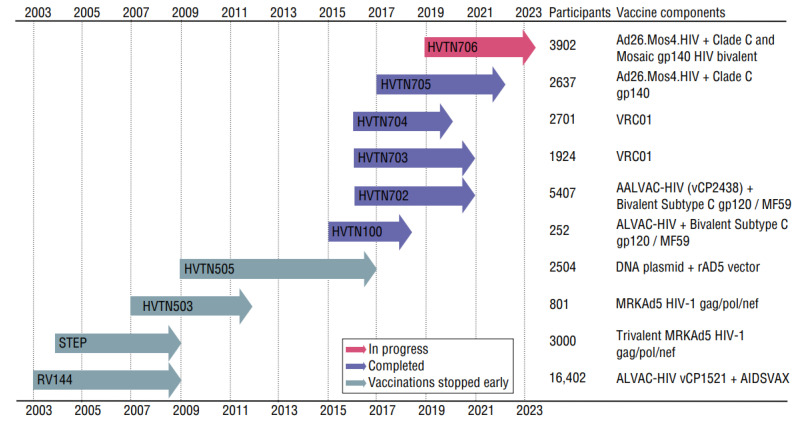
HIV-1 vaccine trial history. Arrows indicate spans of time and are color coded based on their progress. The number of participants and vaccine descriptions are noted. Starting at the left, the **RV144** trial (2003–2006, monitoring until 2009) showed a modest 31% efficacy result [51]. The **STEP** study (2004–2007, monitoring until 2009) found that vaccinated participants were at a higher probability of HIV-1 infection if they had prior high levels of antibodies against the vector, adenovirus serotype 5 [52]. **HVTN503**, also known as the Phambili trial (2007, monitoring until 2012), used a MRKad5 gag/pol/nef HIV-1 subtype B vaccine, similar to the STEP trial. It was stopped within a few months, due to the STEP results. Although similar to the STEP trial, it did not result in an increased risk of HIV infection [53,54]. The **HVTN505** trial (2009–2013, monitoring until 2017) was stopped due to a lack of efficacy. This trial looked at the efficacy of a DNA vaccine with clade B Gag, Pol, and Nef proteins and clade A, B, and C’s Env proteins. There was also a booster administered using a rAd5 vector that expressed clade B Gag-Pol fusion protein and clades A, B, and C’s Env glycoproteins [55]. The **HVTN100** study (2015–2018) was the smallest clinical vaccine trial [56]. **HVTN702**, the Uhambo trial (2016–2021), utilized two priming immunizations of subtype C envelope ALVAC-HIV and four boosters of a subtype C bivalent gp120 protein adjuvanted by MF59 [57]. The **HVTN703**, or AMP trial (2016–2021), tested an intravenous antibody (VRC01). There were ten different infusions over the course of 72 total weeks [54]. In **HVTN704** (2016–2020), AMP participants received the same treatment as the HVTN703 trial [50]. For the **HVTN705**, or Imbokodo trial (2017–2022), a priming vaccination of Ad26.Mos4.HIV was administered at four different time periods with boosting adjuvanted vaccinations of Clade C gp140 protein administered twice [58]. The **HVTN706** or Mosaico trial (2019–2024, expected) has a similar experimental design to HVTN705. This study administered the priming vaccinations of Ad26.Mos4.HIV at four different time periods with boosting vaccinations of Clade C and Mosaic gp140 protein administered twice [58].

**Figure 5 viruses-15-00107-f005:**
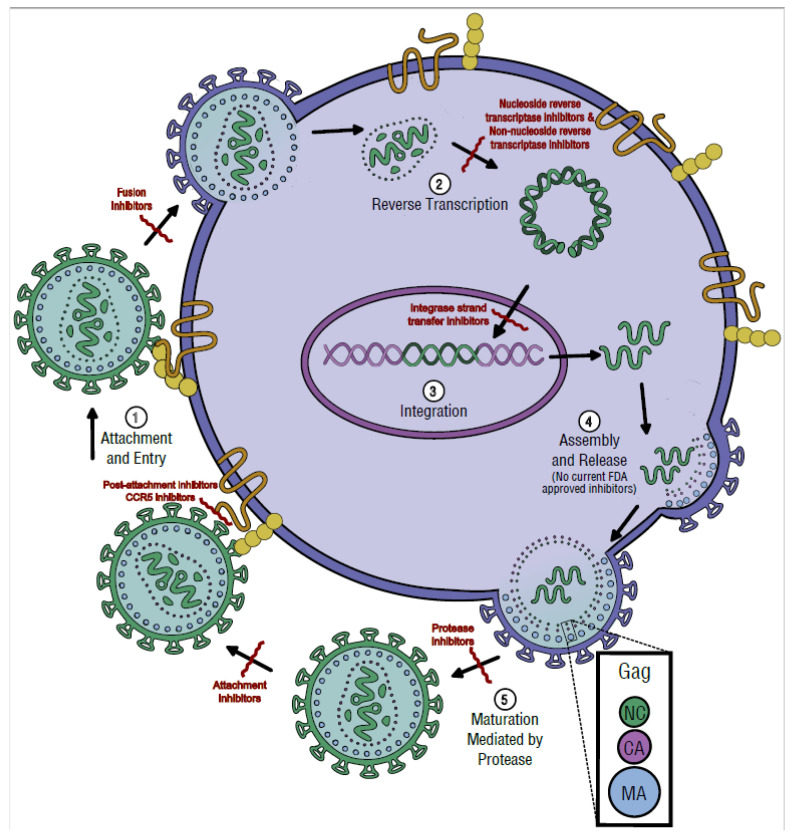
HIV replication cycle and anti-HIV drug targets. **➀**. Attachment begins when HIV’s gp120 spike protein binds to the host cell CD4 receptor and a coreceptor. Membrane fusion is induced, and the viral proteins and genetic content is transferred into the cytoplasm. Several drugs are approved that inhibit these early steps. **➁**. Upon uncoating, the HIV RNA is released and converted to HIV DNA through reverse transcription. Nucleoside reverse transcriptase inhibitors (NRTI) and non-nucleoside reverse transcriptase inhibitors (NNRTI) directly block reverse transcription. **➂**. Integration of viral DNA then occurs. The integrase inhibitor drug class functions as a strand transfer inhibitor (INSTI), which blocks the integrase enzyme from inserting viral DNA into the host DNA. During normal viral replication, the integrated DNA becomes the genomic content that provides the template for viral protein synthesis. New HIV proteins are produced as long polyproteins that are later cleaved by protease into functional enzymes and structural proteins. **➃**. These proteins will move to the cell surface and surround the genome, assembling into immature HIV particles which are released by budding. There is currently no drug able to target the assembly and release of viral particles. **➄**. Inside the released virus, protease acts on the immature HIV particle to create the mature infectious virus. Protease inhibitors (PI) block protease and prevent the formation of mature infectious particles.

**Figure 6 viruses-15-00107-f006:**
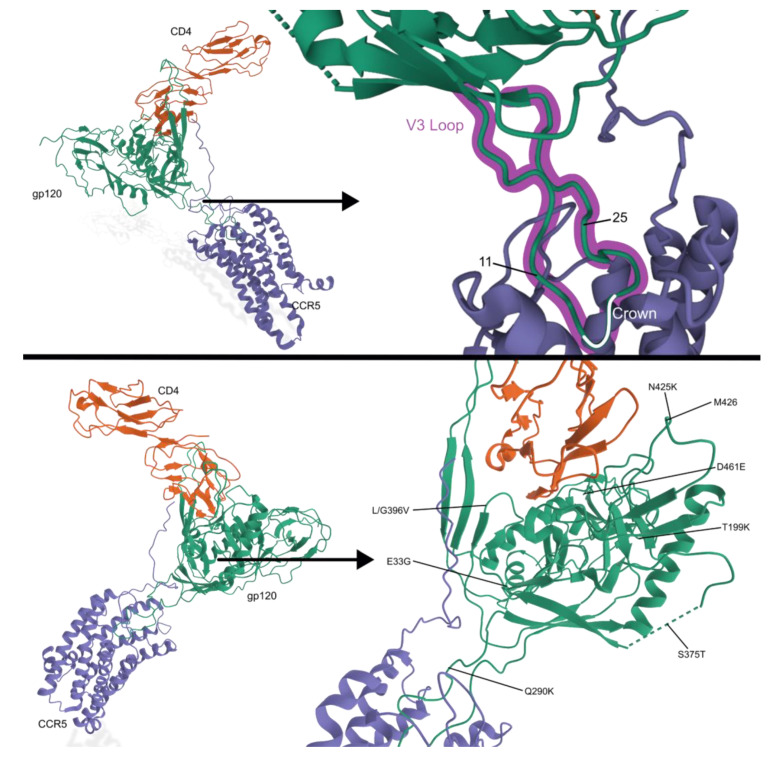
HIV receptor interactions. gp120 (green) interacts with both CD4 (orange) and CCR5 (purple). In the top half, the V3 loop is highlighted in magenta. The 11/25 rule, a common guideline to determine tropism, is associated with the locations noted in the V3 loop. The V3 loop’s crown is a crucial location for binding and is highlighted in white. In the lower half, the C4 region of gp120 is shown with its treatment-associated mutations. Based on PDB 6MEO [45,46,47,96].

**Figure 7 viruses-15-00107-f007:**
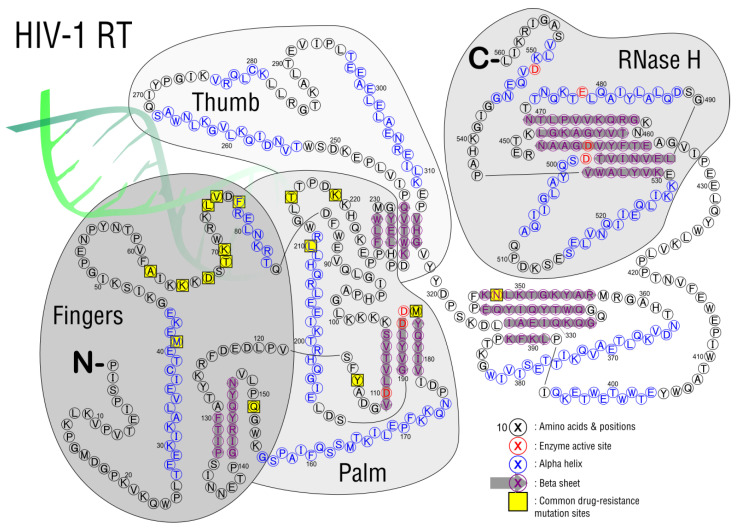
HIV reverse transcriptase enzyme structure in a two-dimensional map, p66 monomer. Mutational hotspots (yellow) are all found in or near the fingers and palm subdomains. In the center of the palm lies the active site residues that catalyze reverse transcription (red). The RNase H domain contains a DEDD box active site that cleaves template RNA [98]. Nucleotide strand position indicated in green.

**Figure 8 viruses-15-00107-f008:**
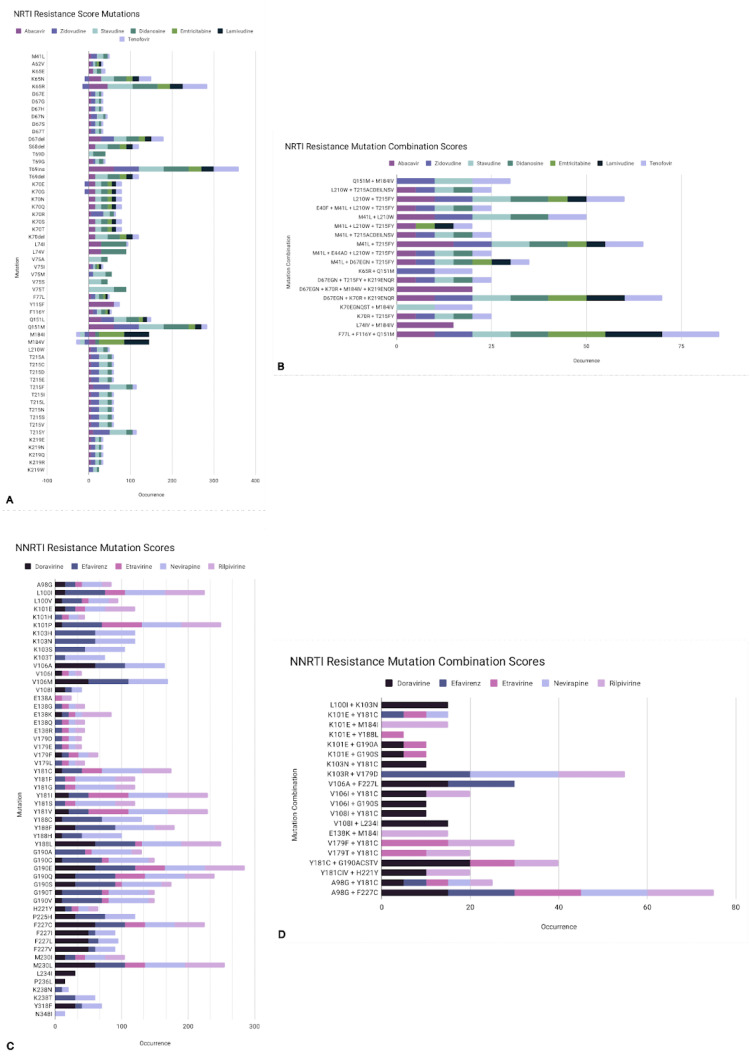
HIV RT-inhibitor resistance mutation scores. These resistance scores, when shown in aggregate, can reflect hotspots in the HIV genome which confers resistance to several drugs. Colors represent different drugs, with occurrence rates combined additively. Any negative value represents that HIV had increased susceptibility (instead of resistance) to that drug. NRTI (**A**) and NNRTI (**C**) mutations are shown. Combinations of mutations are also shown (**B**,**D**) that usually include primary mutations that allow drug escape plus compensatory mutations that reduce fitness costs.

**Figure 9 viruses-15-00107-f009:**
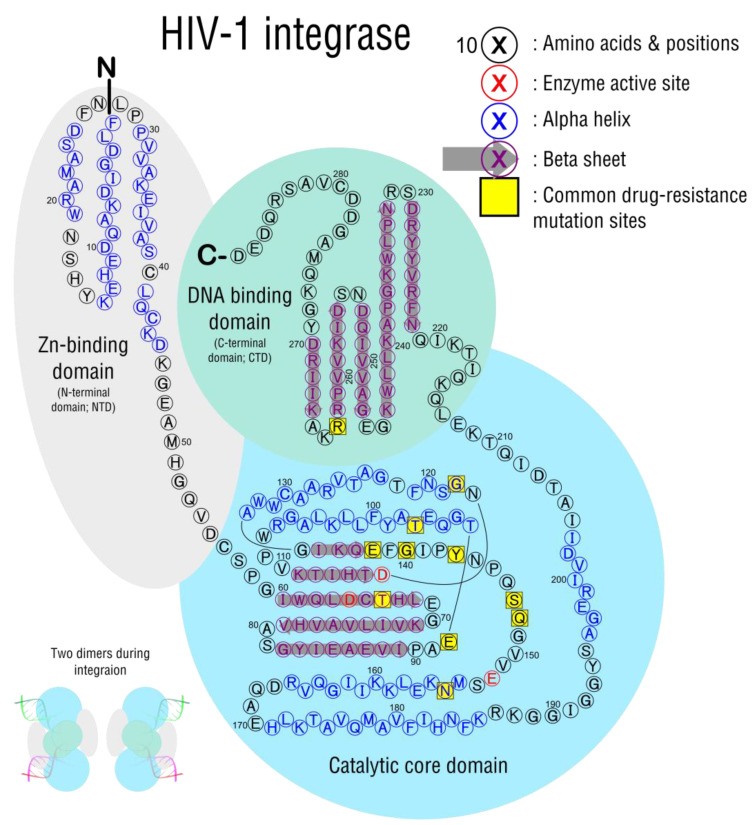
HIV integrase structure, two-dimensional monomer map. Two dimers of this protein bind either end of the HIV genome during integration. The active site is considered to be the essential Asp64, Asp116, and Glu152 (the D,D-35-E motif) [101]. His12, His16, Cys40, and Cys43 bind Zn^2+^ [102]. The sequence of each domain from PDB files 1WJA [103], 1IHV [101], and 1BIS [104]. Multimer structure based on [105]. In the multimer image (lower left), the red DNA (host) and green DNA (virus) represent that these strands will be recombined.

**Figure 10 viruses-15-00107-f010:**
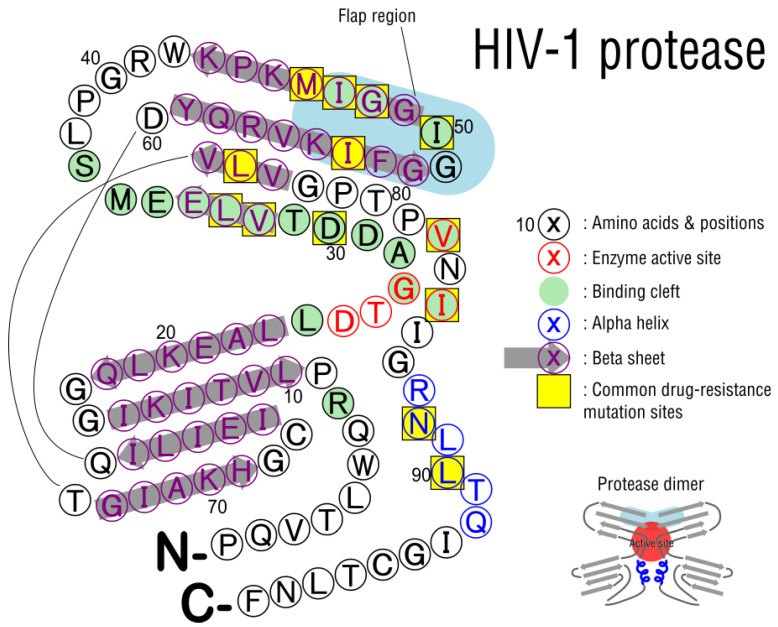
HIV protease structure, two-dimensional monomer map. Protease functions as a dimer and targets various host and viral proteins. HIV polyproteins are cleaved into functional units, most importantly inside maturing capsids. Several drugs interact with the active site and other coordinating amino acids such as I50 in a competitive manner, but mutations may arise that allow for continued function despite drug intervention. Based on sequence from PDB 1AID [134].

**Table 1 viruses-15-00107-t001:** Attachment and entry inhibitors.

Drug Class	Name	Trade Name	Use	FDA Approval
Fusion Inhibitor	Enfuvirtide (T20)	Fuzeon	Patients who are treatment experienced and who currently have virological failure.	13 March 2003
Attachment Inhibitor	Fostemsavir (FTR)	Rukobia	Patients who are treatment experienced, have multi-drug resistant HIV, and who currently have virological failure.	2 July 2020
Post-Attachment Inhibitors	Ibalizumab (IBA)	Trogarzo	Patients who are treatment experienced, have multi-drug resistant HIV, and who currently have virological failure.	6 March 2018
CCR5 Antagonist	Maraviroc (MVC)	Selzentry	Patients who have CCR5-tropic infection, given in combination with other medicines.	6 August 2007

**Table 2 viruses-15-00107-t002:** Mutation patterns that result from attachment and entry inhibitor use.

Name	Significant Mutations	Effect of Mutations
Enfuvirtide	N126K, E137K, S138A	Reduced viral susceptibility and increased viral fusion
	R46K/M on gp41	Potential resistance
	L44M on gp41	Major resistance (1.8 fold)
	V38A/E/K/M on gp41	Increase in CD4 cell count without significant viremia
	Q40H/K/P/T	CD4 loss without significant viremia
	L45Q/M	CD4 loss without significant viremia
	E560K/D/G	Escape peptide inhibitors
Fostemsavir	S375H/I/M/N/T in gp120	Reduced viral susceptibility and may reduce binding of the drug
	M426L/P in gp120	Reduced viral susceptibility and may reduce binding of the drug
	M434I/K in gp120	Reduced viral susceptibility
	M475I in gp120	Reduced viral susceptibility
	L116P	Reduced viral susceptibility
Ibalizumab	N460Q	Disrupts PNGS in V5 loop
	N464Q	Disrupts PNGS in V5 loop
Maraviroc	N425K	May impact CD4 interactions; primary resistance mutation
	E33G, R117Q, Q290K, L/G396V, D461E	May work to stabilize N425K mutation
	T199K in C2 region	Enhances viral fitness
	I318V in V3	Necessary for extensive replication
	F312W, T314A, E317D in V3	Necessary for noncompetitive resistance
	I304V in V3	Enhances viral fitness

**Table 3 viruses-15-00107-t003:** Reverse transcription inhibitors.

Drug Class	Name	Trade Name	Use	FDA Approval
NRTI	Abacavir (ABC)	Ziagen	As part of initial regimen, given in combination with other medicines. Cannot have HLA-B*5701 allele.	17 December 1998
NRTI	Emtricitabine (FTC)	Emtriva	As part of initial regimen, given in combination with other medicines.	2 July 2003
NRTI	Lamivudine (3TC)	Epivir	As part of initial regimen, given in combination with other medicines.	17 November 1995
NRTI	Tenofovir (TDF)	Viread	As part of initial regimen.	26 October 2001
NRTI	Zidovudine (AZT)	Retrovir	No longer used/rarely prescribed.	19 March 1987
NNRTI	Doravirine	Pifeltro	As part of initial regimen, given in combination with other medicines.	30 August 2018
NNRTI	Efavirenz (EFV)	Sustiva	As part of initial regimen, given in combination with other medicines.	17 September 1998
NNRTI	Etravirine (ETR)	Intelence	For treatment-experienced patients, given in combination with other medicines.	18 January 2008
NNRTI	Nevirapine (NVP	Viramune	Given in combination with other medicines.	21 June 1996
NNRTI	Rilpivirine	Edurant	For treatment-naive patients, given in combination with other medicines.	20 May 2011

**Table 4 viruses-15-00107-t004:** Mutation patterns commonly found due to NRTI and NNRTI use.

Mutation	Drug Inhibition	Response/Resistance	Increase Susceptibility
M184V/I	3TC, FTC	Reduces viral replication in vitro and in vivo	AZT, d4T, TDF
K65R	TDF, ABC, ddl, 3TC, FTC		AZT
M41L and T215Y	AZT, d4T, ABC, ddL, TDF	High resistance: AZT, d4T Low resistance: ABC, ddI, TDF	
D67N	AZT, d4T		
K70R	AZT, d4T, TDF	Intermediate Resistance: AZT Low resistance: d4T, TDF	
L210W, M41L, and T215Y	AZT, d4T, ABC, ddI, TDF	High resistance: AZT, d4T Intermediate Resistance: ABC, ddI, TDF	
T215Y/F	AZT, d4T, ABC, ddI, TDF	Intermediate resistance: AZT, d4T Low resistance: ABC, ddI, TDF	
K219Q/E	AZT, d4T		
K70G/Q/E/T/N/S	D4T, ABC, TDF, 3TC/FTC,	Low resistance: 3TC/FTC	AZT
L74V/I and M184V	ABC, ddl		AZT, TDF
Y115F	ABC, TDF		
G151M with either A62V, V75I, F77L, and F116Y	3TC/FTC, TDF	Intermediate resistance: 3TC/FTC, TDF	
G151M	AZT, d4T, ddI, ABC, 3TC/FTC, TDF	High resistance: AZT, d4T, ddI, ABC Low resistance: 3TC/FTC, TDF	
T69D/N/G	Ddl, d4T		
T69D/N/G and TAMS	AZT		
V75T/M/A/S	D4T, ddl		
N348I	AZT, NVP, EFV	Reduces rate of RNA template degradation	

**Table 5 viruses-15-00107-t005:** Integrase strand-transfer inhibitors. (INSTI).

Drug Class	Name	Trade Name	Use	FDA Approval
INSTI	Cabotegravir (CAB)	Vocabria	For patients who will miss planned cabenuva injection, in combination with rilpivirine, or as short-term pre-exposure prophylaxis.	22 January 2021
INSTI	Dolutegravir (DTG)	Tivicay	As part of initial regimen, given in combination with other medicines.	12 August 2013
INSTI	Raltegravir (RAL)	Isentress	As part of initial regimen; post-exposure prophylaxis.	12 October 2007

**Table 6 viruses-15-00107-t006:** Mutation patterns commonly found due to use of INSTIs.

Significant Mutations	Drug Inhibitions	Response/Resistance
T66A/I/K	BIC, DTG, EVG, RAL	High: EVG Intermediate: RAL Low: DTG
E92Q/G/V	BIC, DTG, EVG, RAL	High: EVG Intermediate: RAL Low: DTG
G118R	BIC, DTG, EVG, RAL	Intermediate: RAL, EVG, DTG Low: BIC
E138KAT with Q148	BIC, DTG, EVG, RAL	High: RAL, EVG Intermediate: DTG
G140SAC with Q148H/R/K	BIC, DTG, EVG, RAL	High: RAL Intermediate DTG
Y143C/R with T97A	RAL	High: RAL
S147G	EVG	Intermediate: EVG
Q148H/R/K/N with G140S/A or E138K	BIC, DTG, EVG, RAL	High: RAL, EVG Intermediate: DTG, BIC
N155H/S/T/D	BIC, DTG, EVG, RAL	High: EVG Intermediate: RAL
R263K	BIC, DTG, EVG, RAL	Intermediate: BIC, DTG, EVG

**Table 7 viruses-15-00107-t007:** Protease inhibitors.

Drug Class	Name	Trade Name	Use	FDA Approval
PI	Atazanavir (ATV)	Reyataz	Given in combination with other medicines.	20 June 2003
PI	Darunavir (DRV)	Prezista	Given in combination with other medicines.	23 June 2006
PI	Fosamprenavir	Lexiva	Given in combination with other medicines.	20 October 2003
PI	Ritonavir (RTV)	Norvir	Given in combination with other medicines.	1 March 1996
PI	Tipranavir (TPV)	Aptivus	For treatment-experienced patients, given in combination with other medicines.	22 June 2005

**Table 8 viruses-15-00107-t008:** Protease inhibitor mutation patterns that result due to PI use.

Mutations	Drug Inhibitions	Response/Resistance	Increased Susceptibility
D30N	NFV	High: NFV	N/A
V32I with I47V/A	IDV, FPV, LPV, DRV, ATV, LPV, NFV, TPV	High: LPV Intermediate: DRV	N/A
L33F with other PI-resistance mutations	DRV, FPV, LPV, NFV, TPV	N/A	N/A
M46I/L/V alone or in combination with V32I, I47V, I84V, L90M, I54V, or V82A	IDV, NFV, FPV, ATV, LPV, TPV	N/A	N/A
I47V/A alone or with V32I	IDV, FPV, LPV, DRV, NFV, TPV	High: LPV, FPV Low/intermediate: IDV, DRV, NFV, TPV	SQV
G48V/M/A/S/T/Q	SQV, IDV, LPV, NFV, ATV	High: SQV Intermediate: ATV Low: NFV, IDV, LPV	N/A
I50V/L	FPV, LPV, DRV, ATV	High: ATV	TPV
I54V/T/A/L/M	IDV, LPV, FPV, DRV, ATV, NFV, SQV, TPV	N/A	N/A
L76V	IDV, LPV, DRV, FPV	N/A	ATV, SQV, TPV
V82A/T/S/F/L/M/C	IDV, LPV, ATV, NFV, SQV, FPV, TPV, DRV	N/A	N/A
I84V/A/C	IDV, LPV, DRV, SQV, TPV, FPV, NFV, ATV	High: IDV, LPV, DRV, SQV, TPV, FPV, NFV, ATV	N/A
N88DS	NFV, ATV, IDV, SQV	N/A	FPV
L90M	SQV, NFV, IDV, LPV, ATV, FPV	N/A	N/A

## Data Availability

No new data were created or analyzed in this study. Data sharing is not applicable to this article.

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
