# Peer review of "The Effect of Treatment-Associated Mutations on HIV Replication and Transmission Cycles"

_viruses, 2022, doi:10.3390/v15010107_

Round 1

Reviewer 1 Report

Manuscript title : The effect of treatment-associated mutations on HIV replication and transmission cycles. Overall, this article is interesting and well-written. The authors define and pose questions of research well and interestingly. They choose the right methods and tools to find the answer to that questions. And above all, able to answer questions clearly and concretely.  I appreciate this manuscript to publish in the journal.

The sequence of the content presentation can be done in detail and with beautiful images. It is a system for presenting understanding from empirical evidence.

- Conclusions There should be an expanded discussion of the concluding points. however, This may lead to recommendations for further research in the future. or policy recommendations from the presentation of the results of this research to the agency Related organizations on how each sector should participate, and do what in practice to achieve the best results.

Author Response

Reviewer #1 Comments and Suggestions for Authors

  • Manuscript title: The effect of treatment-associated mutations on HIV replication and transmission cycles. Overall, this article is interesting and well-written. The authors define and pose questions of research well and interestingly. They choose the right methods and tools to find the answer to that questions. And above all, able to answer questions clearly and concretely.  I appreciate this manuscript to publish in the journal.
    • The authors are pleased to contribute the article for publication
  • The sequence of the content presentation can be done in detail and with beautiful images. It is a system for presenting understanding from empirical evidence.
    • To clarify captions, several “caption titles” were added or altered (see captions for Figs. 4 – 10)
  • Conclusions There should be an expanded discussion of the concluding points. However, This may lead to recommendations for further research in the future. or policy recommendations from the presentation of the results of this research to the agency/Related organizations on how each sector should participate, and do what in practice to achieve the best results.
    • The authors were able to moderately expand the final paragraph of the conclusion statement, which now reads, “But “multiple targets” could also describe the coordinated efforts of government, pharmaceutical, scientific, and healthcare agencies, along with the community and the individual. To increase awareness, campaigns can provide education, testing, and treatment. When HIV-positive status is known, immediate access to treatment and education is critical. Monitoring overall patient health, not just HIV progression, will improve health outcomes. These testing and monitoring efforts can inform patient care for patients and their sexual or needle-sharing contacts over time. Because knowledge is power, sharing risk data, care data, and treatment best practices can be expanded and standardized. Databases of mutations and sequencing efforts have reduced time wasted on ineffective treatments in more developed countries, but individuals in low and middle-income countries lack the resources to keep up with effective treatment plans, which can lead to an increase in transmitted-drug resistance. Research will continue to support new therapeutics, preventatives, and vaccine candidates, funded by many levels of philanthropy and government investment. Addressing misinformation and healthcare mistrust will continue to be important. These multifaceted strategies can match the overwhelming speed of viral replication and the subsequent mutations. Coordinated efforts are required at many levels to achieve successful results, but will eventually lead to a cure.”

Reviewer 2 Report

The manuscript by Johnson and colleagues is a nicely written review that covers most aspects of HIV mutations.

What I am missing is a bit more about how genotypic resistance testing is done these days. I mean for example the usage of online interpretation tools. Second, viral sequences archived in the reservoirs is only shortly mentioned. However, the impact of viruses transmitted with drug resistance mutations that have a significant fitness loss (e.g. K65R) is not well discussed. Even so these viruses can quickly become undetectable, they can reemerge from the reservoir if the therapy regime includes the antiviral (e.g. tenofovir) and lead to treatment failure.

Some minor critics:

The reference numbers are sometimes placed before and sometimes behind the period.

Page 3, first paragraph:  The sentence “Non-adherence to the treatment typically correletes with virological failure that lacks drug resistant mutations. Is misleading in my understanding. Clarification is needed.

Page 6, 2.3.1.: Whether broadly neutralizing antibodies bind to the gp160 is irrelevant. They bind to gp120 or gp41 in the trimer.

Page 19, 3.3 Integration: Integration happens with all retroviruses, not only orthoretroviruses. spumaretroviruses do also integrate.

Author Response

  • The manuscript by Johnson and colleagues is a nicely written review that covers most aspects of HIV mutations.
    • Thank you for your critical reading and insights
  • What I am missing is a bit more about how genotypic resistance testing is done these days. I mean for example the usage of online interpretation tools.
    • A new section was added, section 2.2. Drug Resistance Testing that includes more information about resistance testing

2.2. Drug Resistance Testing

Drug resistance testing is necessary to identify the mutations, properly treat the patient, and prevent further drug-resistant mutations. After viral escape, the source of drug resistance can be identified with genotypic testing of HIV reverse transcriptase, integrase, or protease sequences.

When either acquired or transmitted mutation happens, genotypic testing uses Sanger sequencing to identify specific mutations, and to determine an alternative treatment plan. Due to unreliable detection of low-abundance variants, next generation genotype sequencing is gaining popularity, becoming more readily available in terms of cost and accessibility, mostly in developed countries [15,16]. This type of testing requires nucleic acid extraction, PCR amplification, library preparation, sequencing, and data analysis.

Once data is obtained, test interpretation begins. The basic process includes comparing a patient’s resistance mutation to known patterns, and then changing the drug regimen if necessary. Early experiments first discovered these resistance patterns in the genome of HIV [20], and then tested outcomes based on switching to new therapy versus placebo [21,22]. However, not all mutations require a change of regimen, for example when a single component exhibits only low level resistance, but the other components of the drug regimen remain active.

It is clear that mutation patterns in drug-induced cases are nonpolymorphic—they occur in defined patterns not seen in treatment naïve patients, who have more polymorphic natural mutations. It can thus be assumed that if a patient exhibits virological escape (increased serum viral titers) and common drug-resistance mutations are present, it is directly due to the currently prescribed drug. A change in drug regimen may be indicated.

The Stanford HIV Drug Resistance Database (HIVDB, http://hivdb.stanford.edu) [4] allows input of user data and displays penalty scores that should be used to guide clinical decisions. By comparing a patient’s sequence results to databases of mutation patterns, other drug options can be considered. Each potential drug is scored for its susceptibility or resistance to a patient’s drug-resistant strain.

In contrast to genotypic testing, phenotypic testing is used to observe the replicative abilities of an individual’s HIV in the presence of different ARV drugs by molecular cloning of PCR-amplified fragments from clinical samples into cell culture infection systems. Then, exposure to a variety of ARVs can determine the effectiveness of a drug on the individual’s strain. This provides important information to guide an effective treatment regimen, but phenotypic testing is reserved for drug research or perplexing cases

  • Second, viral sequences archived in the reservoirs is only shortly mentioned. However, the impact of viruses transmitted with drug resistance mutations that have a significant fitness loss (e.g. K65R) is not well discussed. Even though these viruses can quickly become undetectable, they can reemerge from the reservoir if the therapy regimen includes the antiviral (e.g. tenofovir) and lead to treatment failure.
    • This paragraph was expanded: “Transmitted drug resistance (TDR) occurs when an uninfected person acquires a strain of HIV that is already resistant to antiretroviral drugs (Figure 1, blue). The ultimate source of transmitted drug resistance is always from another person who has undergone treatment and has an acquired-drug-resistant virus. However, if rounds of “onward transmission” occur with a drug resistant mutant, these strains are more likely to have mutations with reduced viral fitness [1]. The replication of a low-fitness-cost mutation is favorable for the virus when compared to high fitness cost: the lower the fitness cost of a mutation, the more likely it will remain over time, even after those drugs are not used for a patient’s treatment. On a similar note, mutants with high fitness cost can still be transmitted, especially from treatment-experienced patients [12]. These low abundance variants may remain over time until treatment is initiated in the new host, whereupon these resistant strains are selected for quickly, despite the fitness cost. These low abundance variants may be difficult to detect, but can have strong clinical effects [13]. For example, the common K65R mutation may exist at low levels in TDR cases. These pre-existing K65R mutants would then expand to become the dominant variant in a host initiating treatment, such as with tenofovir [14].

  • Some minor critics:
    • The reference numbers are sometimes placed before and sometimes behind the period.
        • This has been fixed
    • Page 3, first paragraph:  The sentence “Non-adherence to the treatment typically correlates with virological failure that lacks drug resistant mutations. Is misleading in my understanding. Clarification is needed.
      • This has been changed to: Non-adherence to the treatment typically correlates with virological failure due to drug resistant mutations. Strictly following the current treatment plan will reduce this possibility because the mutated population will be sufficiently suppressed.
    • Page 6, 2.3.1.: Whether broadly neutralizing antibodies bind to the gp160 is irrelevant. They bind to gp120 or gp41 in the trimer.
      • Changed to: Broadly neutralizing antibodies (bnAbs) bind to gp120 or gp41 and have the potential to match the mutational speed of HIV, maintaining inhibition of attachment to CD4 receptors.
    • Page 19, 3.3 Integration: Integration happens with all retroviruses, not only orthoretroviruses. spumaretroviruses do also integrate
      • Changed to: Integration is a necessary step in the viral replication cycle that happens in all RNA retroviruses [94] and is catalyzed by the viral integrase enzyme (IN).